



# Multiple hazards and risk perceptions over time:
# The availability heuristic in Italy and Sweden under COVID-19

Giuliano Di Baldassarre[1], Elena Mondino[1], Maria Rusca[1], Emanuele Del Giudice[2], Johanna Mård[1], Elena

Ridolfi[1], Anna Scolobig[3,4] and Elena Raffetti[5]

[1] Centre of Natural Hazards and Disaster Science, Department of Earth Sciences, Uppsala University, Uppsala, Sweden
[2] Psychiatry Northwest, Region Stockholm, Stockholm, Sweden
[3] Environmental Governance and Territorial Development Institute, University of Geneva, Switzerland
[4] International Institute for Applied Systems Analysis (IIASA), Vienna, Austria
[5] Department of Global Public Health, Karolinska Institutet, Stockholm, Sweden

*Correspondence to*: Giuliano Di Baldassarre (giuliano.dibaldassarre@geo.uu.se)

**Abstract.** The severe impact of global crises, such as COVID-19 and climate change, is plausibly reshaping the way in which people perceive risks. In this paper, we examine and compare how global crises and local disasters influence public perceptions

of multiple hazards in Italy and Sweden. To this end, we integrate information about the occurrence of hazardous events with the results of two nationwide surveys. These included more than 4,000 participants and were conducted in two different phases of the COVID-19 pandemic corresponding to low (August 2020) and high (November 2020) levels of infection rates. We found that, in both countries, people are more worried about risks related to experienced events. This is in line with the cognitive process known as availability heuristic: individuals assess the risk associated with a given hazard based on how

easily it comes to their mind. Epidemics, for example, are perceived as less likely and more impactful in Italy compared to Sweden. This outcome can be explained by cross-country differences in the impact of, and governmental responses to, COVID-19. Notwithstanding the ongoing pandemic, people in both Italy and Sweden are highly concerned about climate change and they rank it as the most likely threat.

## 1 Introduction

The COVID-19 pandemic is a global concern (Hsiang et al., 2020; Baker et al., 2020). In addition to infections and fatalities (Scudellari, 2020), indirect effects of the ongoing pandemic include severe economic crises, increasing poverty, and exacerbating social inequalities (Nicola et al., 2020; Burki, 2020). Moreover, a deterioration of mental health has been observed among the general population (Sher, 2020), stress- and trauma-related disorders (Thakur and Jain, 2020), mood disorders (Mucci et al., 2020), and domestic violence (Mazza et al., 2020). School closures affected up to 1.6 billion students worldwide

(UNESCO, 2020). Prolonged school closure is believed to have had negative impacts on the well-being and education of



children, and on child labour, teenage pregnancies and persisting socioeconomic and gender disparities, as well as on society at large (Lee, 2020; UNESCO, 2020).

Concurrently, humanity is facing climate change. Storms, floods, droughts and wildfires severely affect many countries around the world with increasing frequency or severity (Balch et al., 2020; IPCC, 2012). In 2019, more than 11,000 scientists declared

"clearly and unequivocally that planet Earth is facing a climate emergency" (Ripple et al., 2020). The United Nations Office for Disaster Risk Reduction recently published an updated report about the Human Cost of Disasters (UNDRR, 2020), showing that "extreme weather events have come to dominate the disaster landscape in the 21$^{st}$ century"(UNDRR, 2020).

The severe impacts of global crises, such as COVID-19 and climate change, have plausibly influenced how people characterise and assess multiple hazards. At the same time, the occurrence of these global crises provides a window of opportunity for

change towards reducing vulnerabilities, while promoting physical, mental and social well-being (Brundiers and Eakin, 2018; Adger et al., 2013; Blumenthal et al., 2020). Thus, understanding public risk perception can contribute to develop policy for desired social transformations, including the protection and improvement of public health, disaster risk reduction and climate change adaptation (Aerts et al., 2018; Buchecker et al., 2013; Dryhurst et al., 2020; Erev et al., 2020; Lee et al., 2015; Marquart-Pyatt et al., 2014; Poortinga et al., 2019; Schneider et al., 2021; Slovic, 1987a; Smith and Mayer, 2018; White et al., 1978;

Bubeck et al., 2012).

A large body of research has shown that the way in which people think about risk depends on emotional, cognitive and cultural factors (van der Linden, 2017) along with levels of media coverage (Kasperson et al., 2016), trust (Terpstra, 2011), knowledge (Mondino et al., 2020b), and experience (Wachinger et al., 2013). A direct experience of an event, in particular, provides an illustration of the threat and demonstrates its potential for future risk (Wachinger et al., 2013). Thus, disasters and crises often

influence public risk perception, as many people internalize the experienced event as a more likely and impactful risk for the future. A critical role in this process can be played by the availability heuristic (Tversky and Kahneman, 1973; Pachur et al., 2012; Sunstein, 2006), as people tend to assess risks based on the ease with which examples of harm come to mind.

In this paper, we compare public perceptions of multiple hazards in Italy and Sweden during the COVID-19 pandemic, and explore whether the availability heuristic can explain cross-country differences. To this end, we integrate information about

the occurrence of hazardous events with the results of two nationwide surveys. These included more than 4,000 participants and were conducted in two different phases of the COVID-19 pandemic corresponding to low (August 2020) and high (November 2020) levels of infection rates in both countries. Similarities and differences between Italy and Sweden allow us to investigate the role played by experience. The way in which people think about epidemics, for example, is expected to have been substantially influenced by COVID-19, which has severely affected both countries, but to which the Italian and Swedish

authorities responded differently.



## 2 Data and methods

### 2.1 Occurrence of disasters

To consistently compare the occurrence of disasters in Italy and Sweden, we used the global archive EM-DAT developed by
the Centre for Research on the Epidemiology of Disasters (EM-DAT: The CRED/OFDA International Disaster Database). A
given event is recorded as a disaster into the EM-DAT database only if at least one of the following criteria is fulfilled: i) 10
or more casualties; ii) 100 or more people affected/injured/homeless; iii) declaration by the country of a state of emergency
and/or an appeal for international assistance.

The EM-DAT database is one of the world's most comprehensive disaster databases and a recent study showed its data was
consistent with the insurance group Munich RE's NatCatSERVICE database (Formetta and Feyen, 2019), but it is nonetheless
subject to limitations. There is some missing information (Voss and Wagner, 2010), and spatial discrepancies resulting from
changes in political boundaries (Gall et al., 2009). Yet, the former issue is mainly related to data before the 1970s, which were
not considered in our analysis, while the latter issue is not affecting Italy and Sweden as their political boundaries have
remained unchanged.

### 2.2 National surveys of public risk perception

To assess public risk perceptions in Italy and Sweden, we performed two nationwide surveys in 5-19 August 2020 and 9-25
November 2020 (Mondino et al., 2020a). These periods correspond to two different phases of the COVID-19 pandemic with
low and high levels of infection rates and excess mortality, as depicted in Figure 1.

The two national samples used in the surveys are considered representative of the Swedish and Italian population (Mondino et
al., 2020a). A total of 4,154 individuals participated in August 2020 (2033 in Italy and 2121 in Sweden) and 4,168 in November
2020 (2004 in Italy and 2164 in Sweden). Respondents were informed that the participation was voluntary and that they
consented to participate in the study by completing the survey. Our survey was carried out in accordance with the ethical
standards set by the European Union under Horizon 2020 (EU General Data Protection Regulation and FAIR Data
Management) and it was approved by the Italian Research Ethics and Bioethics Committee and the Swedish Ethical Review
Authority.

In addition to climate change and epidemics, our survey considered natural hazards directly or indirectly related with climate
change (wildfires, floods and droughts) or with the ongoing COVID-19 pandemic (domestic violence and economic crises).
To investigate the influence of the availability heuristic, we also considered hazards that have recently affected Italy, but not
Sweden (earthquakes) and vice versa (terrorist attacks).

In this study, we focused on three main variables: a) likelihood, b) impact and c) experience. They were derived from the
responses to the three following questions: responses to the three following questions: a) How likely do you think it is that you
are directly involved in [hazard]? b) How much damage do you think [hazard]can cause to yourself? c) Have you ever
experienced [hazard]? Each question was asked in relation to each hazard considered here: epidemics, floods, droughts,

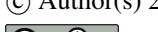



wildfires, earthquakes, terrorist attacks, domestic violence, economic crises, and climate change. Responses were given on a

1-to-5 scale for the first two questions (a, b), while a dichotomic *yes-no* response was used for the last question (c).

For each hazard, national averages of the perceived likelihood and impact were computed from the responses to the first two questions (a, b), while the proportion of people (%) that experienced each type of hazard was derived as a ratio between the number of *yes* responses to the last question (c) and the total number of responses. In addition to national averages, we also considered demographic information provided by the participants –including their age, gender and political orientation– and

explored their role in explaining public perceptions of multiple hazards.

## 3 Results

In this section, we first present the contrasting landscapes of risk in Italy and Sweden by describing the recent occurrence and impact of multiple hazards, including the ongoing COVID-19 pandemic. Based on this, we then compare public risk perceptions in the two countries, and explore the role of the availability heuristic in explaining differences.

### 3.1 Multiple hazards in Italy and Sweden

To compare the risk landscapes in the two countries, we contrasted the number of hazardous events that turned into disasters in Italy and Sweden, according to the global dataset EM-DAT (EM-DAT: The CRED/OFDA International Disaster Database). Figure 2 shows that Italy was affected by numerous disasters associated with natural hazards (see also e.g. Salvati et al., 2016), while Sweden suffered only a few disasters. In the most recent decade (2010-2019), both countries experienced weather-related

disasters. Yet, Italy was severely hit by earthquakes, droughts and flood events, whereas Sweden experienced a catastrophic wildfire (Fig. 2).

Both countries have been severely affected by COVID-19 (Fig. 1). According to the world mortality database (Karlinsky and Kobak, 2021), Italy and Sweden recorded an excess mortality in 2020 of 15.4% and 6.9% respectively. Governmental responses to the pandemic have been different. Italy was the first European country to introduce a national lockdown for over

two months. The Italian response has been primarily driven by its Government via decrees that have introduced (or lifted) stringent national policy response (Paterlini, 2020), including prolonged school closures. In contrast, Sweden drew worldwide attention for its less restrictive measures for fighting COVID-19. The Swedish response has been based on a combination of legally binding rules and general recommendations with heavy reliance on mutual trust between people and authorities (Kavaliunas et al., 2020).

Our survey also considered two indirect effects of the COVID-19 pandemic: economic crises and domestic violence. According to the estimate of the Economist Intelligence Unit, the gross domestic product (GDP) of 2020 shrank by 9.1% in Italy and 3.2% in Sweden with severe outcomes in terms of unemployment and extreme poverty. There have been concerns about increasing domestic violence during the national lockdown in Italy (Lundin et al., 2020). According to the Italian National Institute of Statistics (ISTAT), the number of calls to 1522, the phone number for domestic violence and stalking,





increased by 79.5% in 2020 compared to 2019 (ISTAT, 2021). Yet, robust and comparable data are difficult to find, as the reporting of cases is believed to be incomplete.

Lastly, we examined two additional hazards that have affected the two countries in a different way: earthquakes and terrorist attacks. Since 1980, earthquakes caused a total of 5,419 deaths in Italy and none in Sweden according to EM-DAT. Moreover, while Italy was affected by numerous terrorist attacks from the late 1960s until the late 1980s, the so-called "years of lead",

no major events occurred in the last decades. Instead, a deadly terrorist attack occurred in Stockholm in 2017, one of the most shocking events over the past decade in Sweden (Statement attributable to the Spokesman for the Secretary-General on Attack in Stockholm, Sweden, 2020).

## 3.2 Public risk perception

To compare public perceptions of multiple hazards in Italy and Sweden, we examine the national averages of perceived

likelihood and impact resulting from the two surveys in August and November 2020 (Fig. 3). We found that people in both countries ranked epidemics as one of the most likely hazards. This can be attributed to the salience of the ongoing pandemic and its severe impact in both countries. This outcome is also consistent with recent studies (Dryhurst et al., 2020) that found high levels of epidemic risk perceptions in European countries, including Italy and Sweden. In both countries, public concerns about epidemics increased in the period between August and November 2020 (Fig. 3) plausibly due to the higher levels of

infection rates and excess mortality (Fig. 1).

Overall, epidemics are perceived as less likely but more impactful in Italy compared to Sweden (Fig. 3). As mentioned, Italy responded to COVID-19 with more stringent measures, which have plausibly increased public concerns about the potential negative impact of epidemics. Moreover, the case fatality rate (i.e. deaths per lab confirmed cases) has been substantially higher (about double) in Italy comparted to Sweden throughout the ongoing pandemic (Johns Hopkins Coronavirus Resource

Center, 2021). In terms of indirect effects of COVID-19, Figure 3 shows that economic crises are perceived as both more likely and more impactful in Italy compared to Sweden, which is in line with the fact that Italy's economy was more severely affected by the pandemic.

One striking result is that people in both countries are highly concerned about climate change. This concern can be partly explained by the occurrence of climate-related events that turned into disasters: storms in both countries, as well as recent

wildfires in Sweden and numerous floods and droughts in Italy (Fig. 2). The literature suggests that the political environment also plays a role in climate change perceptions (Marquart-Pyatt et al., 2014). Indeed, high levels of public concern were also illustrated by several people taking part in the Fridays For Future movement in both countries in the months before the pandemic, i.e. late 2019.

Furthermore, we found that people in Sweden perceive wildfires as more likely compared to Italy, while people in Italy

perceive floods and droughts as more likely and more impactful compared to Sweden. These results show that public perceptions are consistent with the occurrences of these types of disasters in the two countries, especially the most recent ones: wildfires in Sweden; floods and droughts in Italy (Fig. 2). In the period between August and November 2020, the levels of risk



perception with respect to wildfires and droughts has slightly reduced in both countries. This can be explained by the seasonality of these two hazards, which typically occur in summer. Moreover, we found that the perceived likelihood of floods

has increased in Italy and attribute it to the occurrence of flooding events (including Venice and the Po River) in October 2020. Previous studies showed that public concerns are often very high in the aftermath of a disaster (Slovic, 2000). Yet, they tend to decline and fade away over time (Fanta et al., 2019; Di Baldassarre et al., 2017). Public risk perceptions about terrorist attacks is a case in point. We found that public concerns about terrorism are relatively low in Italy (Fig. 3), where no major events occurred in the last decade. Instead, people in Sweden perceive terrorism as the most impactful threat (Fig. 3), as a

deadly terrorist attack occurred in 2017. In both countries, the perceived likelihood of terrorist attack increased in the period analysed here. The availability heuristic offers one plausible explanation, as two deadly attacks occurred in Europe only a few days before the start of our second survey. The first one in Nice (France) on 29th of October 2020 (3 fatalities) and the second one in Vienna on the 2nd of November 2020 (5 fatalities).

### 3.3 The role of experience

These results suggest a plausible association between the occurrence of hazardous events and public risk perceptions. To further explore the role of experience in explaining public perceptions of multiple hazards, we relate the proportion of people who has indicated in the survey to have experienced each hazard (dichotomic responses yes or no) with the proportion of people who perceived it as likely, i.e. perceived likelihood/impact no less than 4 in a 1-to-5 scale (Fig. 4). Figure 4 shows that perceived likelihood is associated with experience in both countries. The correlation coefficients are relatively high in both

Italy (0.89 in August 2020, and 0.85 in November 2020) and Sweden (0.90 in August 2020, and 0.91 in November 2020). By exploring the role of sociodemographic factors (i.e. gender, age and political orientation), we found that being male, older age, along with having centre-right or right political orientation were generally associated with a lower perceived likelihood and impact of multiple hazards (Fig. S1, S2 and S3 in the Supplementary Material). The only exception was a higher perceived impact of epidemics among elderly, hugely affected in the ongoing pandemic. These outcomes are in line with the risk

perception literature in terms of gender (Galasso et al., 2020; Gustafsod, 1998), age (Weber, 2016) and political orientation (Marquart-Pyatt et al., 2014). Yet, despite these differences in the absolute values of perceived likelihood and impact, the way in which multiple hazards are ranked remain similar across socio-demographic factors (Supplementary Material). Rankings are primarily driven by experience. We interpret this outcome by the major role played by the availability heuristic in explaining cross-country differences in the way in which people perceive and rank multiple hazards.

### 4 Discussion and conclusions


We found that the availability heuristic is an effective analytical lens to explain cross-country differences in terms of public perceptions, and how they change over time. The recent experience of an event is a key determinant of the way in which people assess multiple hazards (Figs. 3 and 4). Yet, the availability heuristic does not operate in an emotional, social and cultural



vacuum. Risk perception also depends on how experience is internalized. Cultural predispositions and social influences also
play a role. Indeed, we found that while cross-country differences in the ranking of multiple hazards is primarily explained by
experience, the magnitude of concerns depends on sociodemographic factors including age, gender and political orientation
(Supplementary Material).

To provide a richer interpretation of our results, we also placed the results of our survey into a global perspective. Public risk
perceptions in Italy and Sweden were compared with two recent surveys about perceptions of: i) scientists from the global
change research community, collected in the 2020 Future Earth's Survey (Garschagen et al., 2020), and ii) decision makers
around the world, described in the 2020 Global Risks Report by the World Economic Forum (World Economic Forum, 2020).
One striking result (see Supplementary Material) is that the relative ranking of perceived likelihood is the same for people in
Italy and Sweden, decision makers, and scientists: climate change (1st), epidemics (2nd) and terrorist attacks (3rd). This result
is fascinating because it shows a countertrend. Indeed, risk perception research has been grounded on the assumption that there
are major differences between risk judgements of scientists and lay people, and that these differences are not due to ignorance
among the public, but are often driven by different concerns (Slovic, 1987; Slovic and Weber, 2002; Starr, 1969). For instance,
by reviewing studies of climate change perceptions in 2010, Weber (Weber, 2010) stated that "citizens' perceptions of the
importance and severity of climate change do not seem to match those of most climate scientists".

Notwithstanding the ongoing pandemic and significant differences in the perception of multiple hazards, people in both Italy
and Sweden are highly concerned about climate change and they rank it as the most likely threat. Moreover, these high public
concerns are close to climate change perceptions of scientists and decision makers. Concurrently, COVID-19 and climate
change can be seen as global crises caused by the unsustainability of human activities (Horton and Horton, 2020). They have
similar underlying causes, and by addressing them, a number of synergies and co-benefits can be generated, as discussed in
the 2020 report of The Lancet Countdown on health and climate change (Watts et al., 2020). Hence, the convergence of people,
decision makers and scientists has the potential to provide public pressure for, and public acceptance of, new investments and
policy change for promoting public health while reducing vulnerabilities to climatic hazards.

**Code availability**

The script to read, process and visualise survey data is freely available at https://github.com/elenamondino/nationwide_survey.
Information on the packages used are listed in the same repository.

**Author contributions**

G.D.B. originally conceived the study. E.M, J.M., E.Ri., M.R. and G.D.B. designed the two nation-wide surveys. G.D.B., E.M.
and E.R. analysed the results of the surveys, while A.S. contributed to the interpretation of the results in terms of risk
perception. E.Ra. contributed to the analysis of the impact of COVID-19, while E.D.G. and J.M. contributed to the analysis of



the response to COVID-19. G.D.B. wrote the first draft of the paper, to which all authors contributed. All authors revised the
final manuscript.

## Competing interests

The authors declare that they have no conflict of interest

## Acknowledgements

This research received funding by the European Research Council (ERC) within the project HydroSocialExtremes: Uncovering
the Mutual Shaping of Hydrological Extremes and Society, Consolidator Grant No. 771678, H2020 Excellent Science.

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


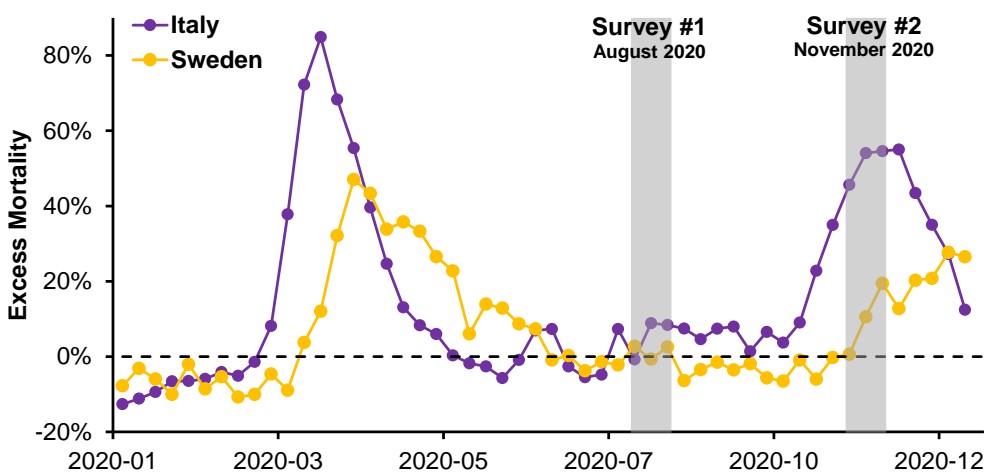

**Figure 1: Nation-wide surveys under the COVID-19 pandemic. Excess mortality data derived from the human mortality database**
**(Karlinsky and Kobak, 2021).**

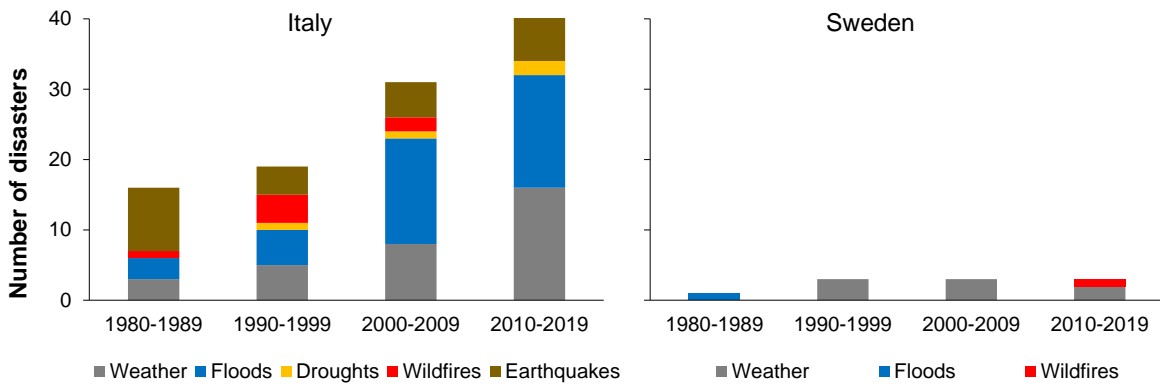

**Figure 2. Contrasting landscapes of risk. Number of naütral hazards that turned into disasters in Italy and Sweden. The label**
**"Weather" is used for meteorological extremes according to EM-DAT terminology, i.e. storms and extreme temperature events**
**(cold/heat waves).**




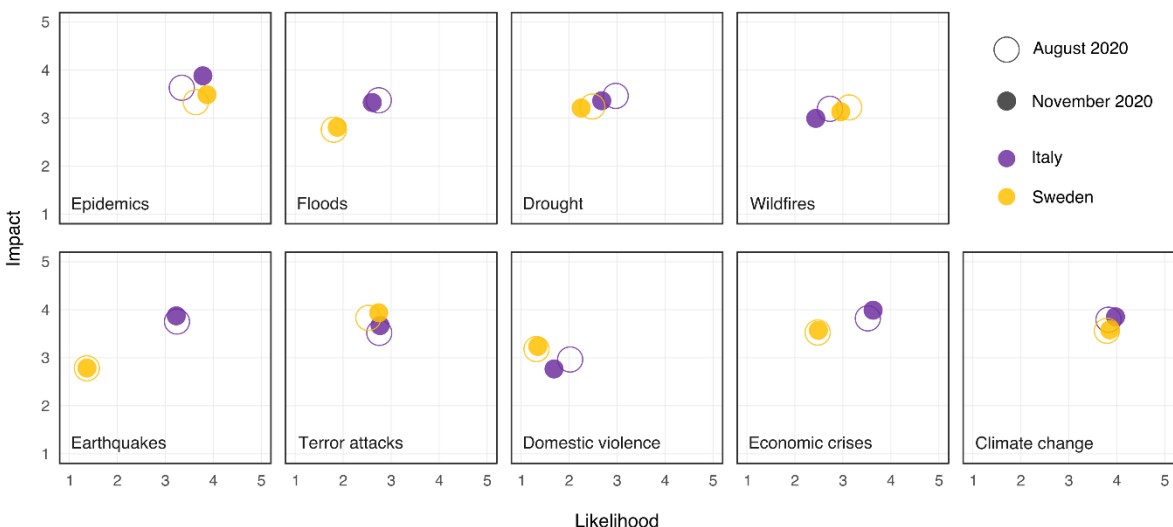

**Figure 3. Perception of multiple risks in Italy and Sweden: National averages of perceived likelihood and impact in August and November 2020.**

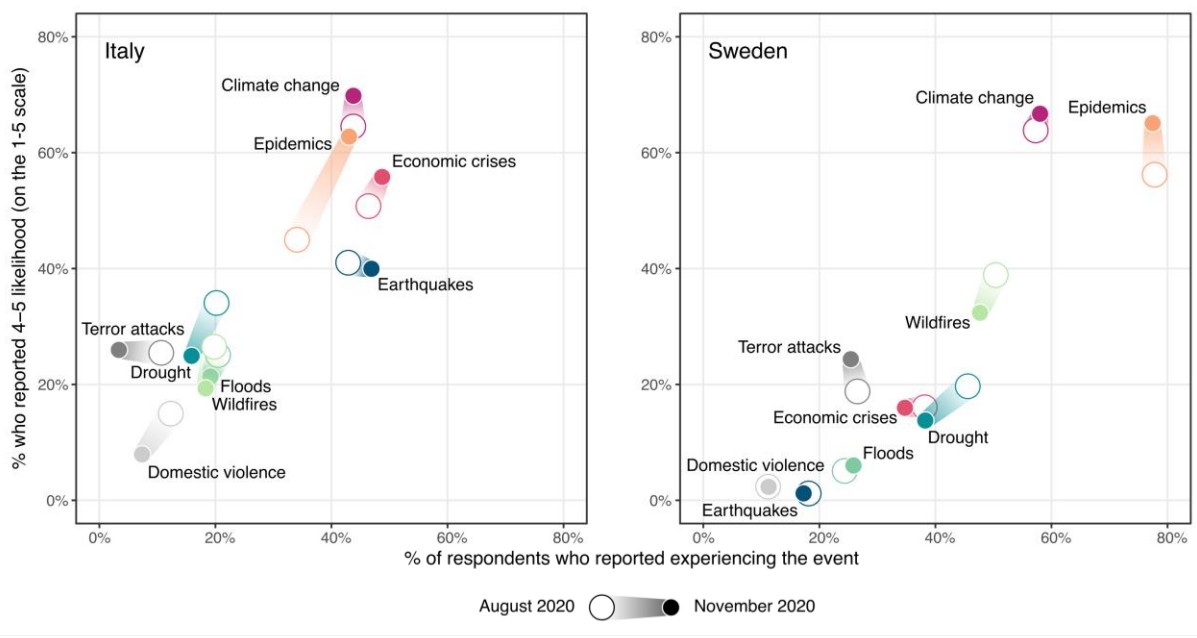

**Figure 4. Experience and perceived likelihood. Proportion (%) of people who have experienced each threat versus % of people who perceived it as likely.**