# Peer review of "Multiple hazards and risk perceptions over time: The availability heuristic in Italy and Sweden under COVID-19"

_Natural Hazards and Earth System Sciences, 2021_

## Author Response (AR1)

**Multiple hazards and risk perceptions over time:**

**The availability heuristic in Italy and Sweden under COVID-19**

**Reply to the Editor**

EDITOR COMMENT: Your manuscript has received very insightful comments from two experts, both suggesting minor revisions. You have provided replies that are adequate within the current stage of the evaluation process. Referees required some additional analysis, but as far as I've understood, when collecting the data with the surveys you were aiming for simple analyses, in favor of "robustness". One issue that was raised from both referees, that I have found very interesting, can be summarized as follows: "How do people experience climate change?". Since "climate" refers to "the weather conditions prevailing in an area over a long period", I am curious about how you have determined that a person knows that he is experiencing climate change. In my humble opinion, people can experience more likely "weather events", which can be sometimes extreme: how do lay people know whether these are due to climate variability (-> weather) or to climate change? Mostly, in this case, the attribution to climate change is given by social/mass media, based on considerations that often differ from more rigorous methods used by scientists. Indeed, the issue of the influence of social media was another point raised by referee #2. I think that in revising their manuscript the authors may better elaborate on this point.

AUTHORS' RESPONSE: We would like to thank the Editor for handling our manuscript, and providing constructive comments to our research work. Along with Referees' comments, these helped us improve and enrich the description of this work. We carefully addressed comments in the revised manuscript, as better specified below in this response letter. We also uploaded an annotated manuscript with track change.

In particular, the revised manuscript addressed the points raised by the Editor. In the Methods section of the revised manuscript, we describe how we people (perceived to have) experienced climate change (see lines 94-99):

In this study, we focused on three main variables: a) likelihood, b) impact and c) experience. They were derived from the responses to the three following questions: a) How likely do you think it is that you are directly involved in [hazard]? b) How much damage do you think [hazard] can cause to yourself? c) Have you ever experienced [hazard]? Each question was asked in relation to each hazard considered here: epidemics, floods, droughts, wildfires, earthquakes, terrorist attacks, domestic violence, economic crises, and climate change.

Indeed, the attribution of extreme events to climate change vs variability is rather complex and often a subject of scientific debate. It is plausible that public perceptions have also been driven by social and mass media, along with the political environment. This aspect, which was also raised by the Referees, is now also discussed in the revised manuscript (see lines 154-160):

One striking result is that people in both countries are seriously concerned about climate change. Indeed, high levels of public concern were also illustrated by several people taking part in the Fridays For Future movement in both countries in the months before the pandemic, i.e. late 2019. This outcome can be partly explained by the occurrence of climate-related events that turned into disasters: storms in both countries, as well as recent wildfires in Sweden and numerous floods and droughts in Italy (Fig. 2). Yet, media are integral to the political environment (Anderson, 2019; Hopke, 2020), which is known to play a major role in climate change perceptions (Marquart-Pyatt et al., 2014). As such, these public concerns can also be attributed to media that have increasingly associated the occurrence of extreme weather events to climate change (Hopke, 2020).

**Response to Reviewer #1**

We would like to thank Anonymous Referee #1 for reviewing our manuscript, being positive about our research, and providing highly constructive comments. These helped us improve the description of this scientific work. We carefully addressed all of them in the revised manuscript, as specified below.

**RESPONSE TO SPECIFIC COMMENTS**

(a) The graphical analysis is interesting and of high quality. Nevertheless, I think that the paper would improve if the authors would examine the data in more detail, for example by a regression analysis. This would allow the authors to analyze the research question while controlling for several variables, such as demographics. Such an analysis could also help to identify which proportion of the variation in risk perceptions can be attributed to different variables, such as objective risks, experience (i.e., availability heuristic) and demographic variables.

(a) We totally agree that these data can potentially be examined in more detail. Yet, some types of detailed analyses are not feasible while considering all hazards. Simple methods are preferred here in order to compare multiple hazards of different nature (for the sake of robustness). More complex models or regression analyses will be used for future studies focusing on specific hazards. See also point (b). We included a new section 2.3 in the revised manuscript discussing the Methods and providing a sound justification for simple approaches. See lines 100-106:

For each hazard, national averages of the perceived likelihood and impact were computed from the responses to the first two questions (a, b), while the proportion of people (%) that experienced each type of hazard was derived as a ratio between the number of yes responses to the last question (c) and the total number of responses. In addition to national averages, we also considered demographic information provided by the participants – including their age, gender and political orientation– and explored their role in explaining public perceptions of multiple hazards. Since this study deals with of multiple hazards of different nature, we kept the methods as simple as possible (e.g. graphical analyses of average values) for the sake of robustness. More complex models or regression analyses will be used for future studies focusing on specific hazards.

(b) The detailed spatial data may allow the authors to examine the difference between objective risks and risk perceptions (risk misperceptions). Which respondents live close to high-risk areas (e.g., high infection levels, or high-risk mountain or potential flooding areas) and does that correspond to the reported risk perceptions?

(b) Indeed, spatially distributed data can be used to examine differences between actual impacts and risk perceptions. Yet, location of respondents is not exact. The survey only provides information about the administrative region (e.g. Tuscany) in which each respondent is located. This prevents us to determine whether they "live close to high-risk areas" for some hazards. In other words, this can be done for hazards operating at larger scale (e.g. droughts), but not for the more localized ones (e.g. floods). Future studies will definitely explore the spatial aspect (also linking to disaster risk reduction policies), but focusing for a sub-set of hazards.

(c) The authors could discuss the findings about climate change a bit further. What disasters were respondents thinking about when answering the question about climate change risks? Is it possible that any overlap exists between 'climate change risks' and 'weather risks'? Would that influence the conclusions of the paper?

(c) We welcome the Referee's suggestion to discuss the findings about climate change. We added a clarification of how we explored risk perceptions (see lines 94-99 of the revised manuscript and above response to the Editor). Yet, we cannot really know "what disasters were respondents thinking about when answering the question about climate change risks". We also added one paragraph in the revised manuscript to discuss the interplay between climate change and occurrence of weather extremes and the potential role of social and mass media (see lines 154-160 of the revised manuscript and above response to the Editor).

(d) Finally, the paper would improve if the authors would discuss consecutive and multirisks. How independent are these risks, and would high risk perception (or worry) for one hazard increase risk perception for another? Some people will have high estimates of the health impacts of epidemics, while others are more concerned about job security in the economic crisis following an epidemic.

(d) We added one paragraph in the revised manuscript addressing this point and referring to the most recent literature in the topic of consecutive and multi-hazards; from line 193 of the revised manuscript:

Over the past decade, scholars in natural hazards have raised the importance of exploring consecutive and multiple risks to inform policies of disaster risk reduction (Ward et al., 2020; Ruiter et al., 2020). In this context, we found that the availability heuristic is an effective analytical lens to explain cross-country differences in terms of public perceptions of multiple hazards, and how they change over time (...)

Lastly, we thank the Referee for providing "technical corrections". We amended them all in the revised manuscript.

**Response to Reviewer #2**

We would like to thank Anonymous Referee #2 for reviewing our manuscript, being positive about the paper, and providing highly constructive comments. These will help us improve the description of this research work. We carefully addressed all of them in the revised manuscript, as specified below.

**RESPONSE TO SPECIFIC COMMENTS**

My first comment is about the broad spectrum of hazards of different nature, frequency and severity the authors analyzed. These hazards require very different prevention (e.g. economic crisis and flood) and preparedness (e.g. earthquake and terrorist attack) measures and they are not homogeneously distributed in the national territory. People living in urban areas are more aware of specific type of hazards (e.g. technological, environmental, criminality) than those they do not experienced or they completely ignore.

The first comment is about the broad spectrum of hazards. We agree with the Referee that they are of "different nature, frequency and severity" and "require very different prevention" "and preparedness measures", while "they are not homogeneously distributed in the national territory". This is exactly why we used simple research methods to compare public perceptions to multiple hazards. More complex models or regression analyses will be used for future studies focusing on specific hazards. The revised manuscript discussed this point (see lines 104-106).

**The geographical distribution of respondents may allow the authors to examine the difference between objective and perceived impacts.**

We also agree with the reviewer that the geographical distribution of respondents can help examine differences between actual impacts and risk perceptions. Yet, this can only be done for a subset of hazards (e.g. droughts) operating at large scale because the exact location of respondents is unknown. The dataset only provides info on the administrative region (e.g. Tuscany). In other words, we don't know who lives, for example, in flood-prone areas. Future studies will address the geographic dimension by focusing on a subset of the hazards where regional scales are meaningful (e.g. droughts).

The knowledge the public has of the different hazards is another important issue. For example, floods are easily recognizable, the domestic violence often remains hidden. Some hazards can be easily related to the season, or the weather conditions, or the geological and geomorphological assets of the territory, others are dependent on variables that people can ignore or not understand (economic crisis).

The third comment relates to the "knowledge the public has of the different hazards". We totally agree with the Referee that this is a key element at play. While this paper focuses on experience, the role of knowledge (that we explored in Mondino et al. 2020b) is explicitly mentioned in the paper:

"A large body of research has shown that the way in which people think about risk depends on emotional, cognitive and cultural factors (van der Linden, 2017) along with levels of media coverage (Kasperson et al., 2016), trust (Terpstra, 2011), knowledge (Mondino et al., 2020b), and experience (Wachinger et al., 2013)."

Another question I would like to ask the authors is: Are people able to distinguish the climate change hazard from floods and drought? How did they experience the climate change? The survey considered natural hazards directly or indirectly related with climate change (wildfires, floods and droughts). Have the authors considered the possible dependence between the hazards and how the relation was handled in the responses analysis?

Excellent questions. We added one paragraph in the revised manuscript to clarify how climate change perceptions were collected and the influence of extreme events (see above, Response to the Editor).

Most of the hazards differently impact the population based on their income, gender, employment, residence, etc. This can have strong implications in how these hazards are perceived across the population and a deeper analysis of the multiple variables can increase the quality of the work.

Indeed, there are multiple factors at play. The role of experience in shaping public perceptions (availability heuristic) is compared with the one of gender, age and political orientation in the Supplementary Material.

A major concern is the global database the authors used to quantify the objective impacts. To compare the occurrence of disasters in Italy and Sweden, they analyzed the global EMDAT archive that is the world's most comprehensive disaster database. The problem in using this type of data is that it lacks systematic information on low to medium intensity and high frequency events. If the heavy impact of a low frequency disaster can modify the public perception, how does the public respond to minor, but extremely frequent and not recorded in the global database, damaging events? Can this gap influence the proportion between the perceived impacts and the likelihood of occurrence? The hazard classification used in the EMDAT is quite different from the list of hazard the authors investigate. How did they handle with this mismatch?

The Referee discuss the limitations of EM-DAT. It should be noted that we never used EM-DAT data for quantitative comparisons across hazards. They are only used to introduce the two case studies (Italy and Sweden) and contrast their risk landscapes. Still, we revised the manuscript by complementing the paragraph about the limitation of the dataset (see lines 69-76)

The EM-DAT database is one of the world's most comprehensive disaster databases and a recent study showed its data was consistent with the insurance group Munich RE's NatCatSERVICE database (Formetta and Feyen, 2019), but it is nonetheless subject to limitations. There is some missing information (Voss and Wagner, 2010), and spatial discrepancies resulting from changes in political boundaries (Gall et al., 2009). Yet, the former issue is mainly related to data before the 1970s, which were not considered in our analysis, while the latter issue is not affecting Italy and Sweden as their political boundaries have remained unchanged. Moreover, EM-DAT does not capture minor events that can be extremely frequent, such as wildfires in Sweden. For all these reasons, information about the occurrence of disasters was only used to contrast (qualitatively) the risk landscapes in Sweden and Italy.

I would also like to briefly mention the issue of sharing experiences through new digital mode, so current under covid-19. Tools such as social media can have influenced the answers on the impacts and on the likelihood. In recent years even during minor damaging events (natural and non-natural) images and videos, from social media and news reports, quickly reach the widest audience leaving a trace. Even though they did not have direct experience of the damaging event, they share it through images, videos and stories. Can this influence the public perception of the actual impacts?

Indeed, media have plausibly influenced public risk perception. We revised the manuscript to more explicitly discuss their role, especially for climate change perceptions (see lines 154-160 of the revised manuscript and above response to the Editor).

To provide a richer interpretation of the results for three of the numerous hazards investigated, the authors compared the results of their surveys with two recent surveys about perceptions of scientists, collected in the 2020 Future Earth's Survey, and of decision makers, described in the 2020 Global Risks Report by the World Economic Forum. The results are reported in the supplementary material. If I centered the importance of the matter, in the comparison there are no major differences between risk

judgements of scientists (from the two recent reports) and lay people (from the two surveys). I think it could be very interesting to deepen this point of discussion adding possible reasons for their findings inside the paper and not in the supplementary materials.

We followed the Referee's suggestion of including in our main text the comparison of our surveys with two recent surveys about perceptions of scientists, collected in the 2020 Future Earth's Survey, and of decision makers, described in the 2020 Global Risks Report by the World Economic Forum. See revised manuscript (lines 202-212) and new Fig. 5.

---

## Author Response (AR2)

**Response to the Editor**

*Dear Authors,*

*thank you for your revised manuscript. You have taken into account the most relevant issues raised by the referees, as well as my comments.*

We thank the Editor for providing minor comments that helped us further improve the manuscript. We have addressed all of them as follows:

*- Perhaps you can improve the introduction at LL 33-37 by adding a short statement regarding the issue about the interaction between media and climate change perception and the related uncertainty.*

We added a sentence about the increased media coverage of climate issues in the introduction. Lines 37-39 of the revised manuscript:

"Furthermore, media coverage of climate issues has increased in many regions of the world over the past years (Hopke, 2020)."

Perceptions here are only introduced after line 48, with reference to media:

"A large body of research has shown that the way in which people think about risk depends on emotional, cognitive and cultural factors (van der Linden, 2017) along with levels of media coverage (Kasperson et al., 2016)…"

*- If the data are easly available, it would be perhaps more significant to compare (LL 130) the number of calls in 2020 with the average yearly value in a period of multiple years, instead to the value of 2019 alone.*

Data are available only for 2018, 2019 and 2020 (https://wwwistat.it/it/files//2021/05/Case-rifugio-CAV-e-1522.pdf). As suggested, we provided % change with respect to the average yearly value in the period 2018-2019:

"According to the Italian National Institute of Statistics (ISTAT), the number of calls to 1522, the phone number for domestic violence and stalking, increased by 70,3% in 2020 compared to the average yearly value in the period 2018-2019 (ISTAT, 2021). Yet, robust and comparable data are difficult to find, as the reporting of cases is believed to be incomplete."

*- At L 79, when citing Mondino et al. (2020a), you may add the link to the survey*

*https://static-content.springer.com/esm/art%3A10.1038%2Fs41597-020-00778-7/MediaObjects/41597_2020_778_MOESM1_ESM.pdf*

Thanks for the suggestion. We added text and link in line 89 of the revised manuscript:

"The questionnaires used in the national surveys are freely available online at:

https://static-content.springer.com/esm/art%3A10.1038%2Fs41597-020-00778-7/MediaObjects/41597_2020_778_MOESM1_ESM.pdf ."

*- At L 160 before "media" a "to" seems missing.*
Amended.